# Fabrication and Characterization of Hydrophobic Cellulose Nanofibrils/Silica Nanocomposites with Hexadecyltrimethoxysilane

**DOI:** 10.3390/polym14040833

**Published:** 2022-02-21

**Authors:** Gi-Hong Kim, Dong-Ho Kang, Bich-Nam Jung, Jin-Kie Shim

**Affiliations:** 1Korea Packaging Center, Korea Institute of Industrial Technology, Bucheon 14449, Korea; kakamate@kitech.re.kr (G.-H.K.); kangppp@kitech.re.kr (D.-H.K.); jbn5666@kitech.re.kr (B.-N.J.); 2Department of Chemical and Biological Engineering, Korea University, Seoul 02841, Korea

**Keywords:** cellulose nanofibrils, CNF/silica nanocomposites, hexadecyltrimethoxysilane, superhydrophobicity

## Abstract

Cellulose nanofibrils (CNFs) have attracted much attention because of their renewability and potential biocompatibility. However, CNFs are extremely hydrophilic due to the presence of a large number of hydroxyl groups, limiting their use as a water-resistant material. In this work, we controlled the adsorption behavior of silica nanoparticles on the surface of CNFs by adjusting the synthesis conditions. The silica nanoparticle size and packing efficiency on the CNF surface could be controlled by varying the ammonium hydroxide and water concentrations. In addition, hexadecyltrimethoxysilane (HDTMS) was successfully grafted onto CNF or CNF/silica nanocomposite surfaces, and the quantitative content of organic/inorganic substances in HDTMS was analyzed through XPS and TGA. The HDTMS-modified CNF/silica nanocomposites were more advantageous in terms of hydrophobicity than the HDTMS-modified CNF composites. This is because the silica nanoparticles were adsorbed on the surface of the CNFs, increasing the surface roughness and simultaneously increasing the amount of HDTMS. As a result, the HDTMS-modified CNFs showed a water contact angle (WCA) of ~80°, whereas HDTMS-modified CNF/silica nanocomposites obtained superhydrophobicity, with a WCA of up to ~159°. This study can provide a reference for the expansion of recyclable eco-friendly coating materials via the adsorption of silica nanoparticles and hydrophobic modification of CNF materials.

## 1. Introduction

Environmental issues due to plastics are increasing the demand to replace petroleum-based materials with eco-friendly natural materials. Cellulose, the most abundant resource on earth, has gained substantial attention owing to its renewability, recyclability, biodegradability, nontoxicity, light weight, and good mechanical properties [1,2,3,4,5]. The various types of nanocellulose can be classified according to their width and length; these include cellulose nanocrystals (CNCs), cellulose nanofibrils (CNFs), and cellulose microfibrils (CMFs) [3,6]. CNCs have a short, rod-like shape around 2–20 nm in width and around 100–500 nm in length [7]. CNFs have a long, fibrillar shape around 1–100 nm in width and around 500–2000 nm in length [8,9]. The applications of nanocellulose are limited due to its lack of hydrophobicity, which is caused by the presence of a large number of hydroxyl groups (–OH). Low moisture resistance is one of the most important factors to consider in practical applications, as it seriously affects reliability and performance [10,11,12,13]. Recently, various studies have been conducted on the modification of the surface and structure of nanocellulose. The modification is mainly based on the reactivity of the hydroxyl group for surface and structural properties. It is through functionalization reactions—such as oxidation, etherification, esterification, silylation, and polymer grafting—that nanocellulose can improve its extraction capacity and enhance the hydrophobicity of its surface [7,14,15].

Hydrophobic surfaces are formed due to the combination of two key factors: low surface energy, and surface roughness (i.e., hierarchical structures). A surface can become superhydrophobic when it is imparted with microroughness or nanoroughness. These surface properties are typically characterized by a water contact angle (WCA) greater than 150° and a sliding angle of less than 10° [16,17,18]. This can be explained via the Wenzel state [19] and Cassie state [20], using the “lotus” model. The wetting behavior of surfaces has attracted much attention because of its wide range of applications, such as for self-cleaning, anti-icing, water–oil separation, and anticorrosion. These characteristics are also reported in various forms, such as plastic composites, paints, coatings, packaging films, papers, aerogels, and sponges [21,22,23,24,25,26,27,28,29,30]. Lu et al. [25] reported magnetic/silanized ethyl cellulose sponges prepared from hexadecyltrimethoxysilane (HDTMS) and mixed with ferroferric oxide (Fe_3_O_4_) nanoparticles (WCA > 150°). Feng et al. [26] reported cellulose/silica aerogels developed from recycled cellulose fibers and methoxytrimethylsilane (MTES), with an average WCA of 151°. Baidya et al. [27] reported superhydrophobic CNF composites fabricated from perfluorooctyltriethoxysilane and [3-(2-aminoethylamino)propyl] trimethoxysilane, with a WCA of 160°. Nevertheless, fluorinated compounds are expensive, and may accumulate and become toxic in organisms and the environment [31,32]. In addition, many well-known studies using HDTMS have been conducted to impart roughness and superhydrophobicity to the surface of silica nanoparticles. Some studies have investigated the hydrophobic surfaces of cotton or wood using a HDTMS-modified silica nanoparticle coating solution [33,34,35,36,37,38]. Although many previous studies have focused on the phenomenon of imparting hydrophobicity, quantitative analysis of the modification of HDTMS does not provide a sufficient explanation.

In this study, hydrophobic CNF/silica nanocomposites were prepared through the adsorption of silica nanoparticles on the CNFs’ surface and subsequent modification using HDTMS (Figure 1a). We attempted to impart hydrophobicity using HDTMS with a long alkyl group, and precursors such as MTES with a relatively short alkyl group were excluded from the selection because the formation of nanoparticles was dominant [39]. The CNF/silica nanocomposites were prepared via the sol–gel method. The adsorption behavior and silica nanoparticles could be controlled by adjusting the synthesis conditions, such as water and ammonium hydroxide concentrations, and the scale-up process was applied. The introduction of silica nanoparticles increased the surface roughness, surface area, and thermal stability of the nanocomposites (Figure 1b). The HDTMS-modified CNF/silica nanocomposite could be controlled from around 88° to 159° through the HDTMS concentration (with a HDTMS concentration of around 0.1 to 2.0 wt%). In addition, as a result of the hydrolysis–condensation reaction of HDTMS, it was possible to obtain the contents of the thermally decomposed organic region and the non-decomposed inorganic region through TGA. In particular, it was confirmed that the HDTMS-modified CNF/silica nanocomposite (with a HDTMS concentration of ~2.0 wt%) had an HDTMS content of ~56.18 wt% (inorganic HDTMS 12.24 wt%, organic HDTMS 43.94 wt%), and achieved superhydrophobicity (>159°) in various pH solutions (Figure 1c). The obtained HDTMS-modified CNFs and CNF/silica nanocomposites were characterized via Fourier-transform infrared (FTIR) spectroscopy, X-ray photoelectron spectroscopy (XPS), thermogravimetric analysis (TGA), field-emission scanning electron microscopy (FE-SEM), and transmission electron microscopy (TEM). The effects of HDTMS-modified CNFs and CNF/silica nanocomposites on the morphology, wettability, and roughness were investigated by measuring the water contact angles, and via atomic force microscopy (AFM). The results can provide a reference for the adsorption of silica nanoparticles on CNFs and the hydrophobic modification of CNF/silica nanocomposites, as well as expanding their range of eco-friendly coating applications.

## 2. Materials and Methods

### 2.1. Materials

The CNF dispersion (1 wt%) was purchased from CNNT (Suwon, Korea). The width of the CNFs was in the range of 10–100 nm, and they had a length of several micrometers (Appendix A). The CNFs were manufactured using an aqueous counter collision system, with a fluid containing cellulose fibers that collided with one another at high pressure to fibrillate at the nanoscale. Tetraethyl orthosilicate (TEOS), HDTMS, ammonium hydroxide (NH_4_OH, 28 wt%), and ethanol (CH_3_CH_2_OH) were purchased from Sigma-Aldrich (St. Louis, MO, USA).

### 2.2. Preparation of CNF/Silica, HDTMS-Modified CNF/Silica, and HDTMS-Modified CNF Nanocomposites

In all of these experiments, the CNF dispersion was replaced with ethanol from water. The CNF water dispersion was exchanged via centrifugation dispersion cycles using ethanol. The HDTMS-modified CNF/silica nanocomposites were prepared in two steps by controlling the HDTMS concentration, as shown in Figure 1a. Firstly, the synthesis of the CNF/silica nanocomposite series (CNF/silica-1,2,3, and 4) was examined under different reaction conditions—specifically, by varying the reaction solvent and catalyst concentrations (Table 1). In the first step of the process, ammonium hydroxide was added to the CNF/ethanol solution under stirring and sonication for 15 min. After nitrogen was bubbled at room temperature for at least 30 min to deoxygenate, TEOS was then added at a rate of 2 mL/h. The hydrolysis and condensation reaction was carried out under constant stirring at room temperature for 24 h. After the reaction, the colloid dispersion was purified via centrifugation and redispersion cycles to remove the unreacted precursor and catalyst. Finally, the CNF/silica nanocomposite dispersion was obtained. Secondly, the HDTMS-modified CNF (h-CNF 0.4 and 0.8) and HDTMS-modified CNF/silica nanocomposites (h-sCNF 0.1, 0.5, 1.0, and 2.0) were examined under different HDTMS concentrations (Table 2). HDTMS-modified CNF/silica nanocomposites were prepared using CNF/silica-4 via the sol–gel method at 40 °C. As control samples, HDTMS-modified CNF nanocomposites were also investigated. All samples were purified via several centrifugation and redispersion cycles to remove the unreacted precursor and catalyst. The compositions are shown in Table 1 and Table 2.

### 2.3. FE-SEM and TEM Analysis

The morphology of the CNF, CNF/silica, h-sCNF, and h-CNF nanocomposites was examined via FE-SEM (SU8020, Hitachi, Tokyo, Japan) and TEM (HT7700, Hitachi, Tokyo, Japan). The FE-SEM images were collected at an accelerating voltage of 3 kV. The samples were coated onto a silicon wafer using a solution of CNF/silica nanocomposites diluted in ethanol; the wafer was then coated with Pt/Pd using an ion sputter coater (E-1045, Hitachi, Tokyo, Japan). The TEM images were collected at an acceleration voltage of 100 kV on a carbon-coated copper grid.

### 2.4. Zetasizer Analyzer and Zeta Potential Analysis

The Z-average particle size was measured at 25 °C with a Zetasizer (Nano ZS, Malvern Instruments Ltd., Worcester, UK) using a ~0.1% dispersion solution. The zeta potential (ζ) was determined using a laser electrophoresis zeta potential analyzer (Nano ZS, Malvern Instruments Ltd.). The zeta potential analysis was performed at pH 12 and 25 °C. The zeta potential was calculated from the electrophoretic mobility (*µ*_e_) using Smoluchowski’s equation,
(1)$$\zeta =\frac{4\pi \eta }{\varepsilon }\times \mu_{e}$$
where ζ is the zeta potential in mV, ε is the dielectric constant of the medium, η is the viscosity of the solution, and $\mu_{e}$ is the electrophoretic mobility [40].

### 2.5. FTIR Spectroscopy and XPS Analysis

The chemical structure and composition of the CNF, CNF/silica, h-sCNF, and h-CNF nanocomposites were characterized via FTIR spectroscopy (Cary 600 Series, Agilent Technologies, Santa Clara, CA, USA) in the range of 4000–600 cm^−1^, and via XPS (NEXSA, Thermo Fisher, Waltham, MA, USA) with a monochromated Al *K*α source (1486.6 eV).

### 2.6. TGA

The thermal properties were measured using a thermogravimetric analyzer (Q500, TA Instruments, New Castle, DE, USA). The silica and the inorganic/organic HDTMS were confirmed by the thermal decomposition amount and residual amount. The measurements were performed using a weight of 10 ± 0.5 mg to raise the temperature to 800 °C under nitrogen gas.

### 2.7. WCA and AFM Analysis

The surface properties and WCA were analyzed using a contact angle analyzer (SmartDrop standard, FEMTOFAB, Seongnam, Korea); the measurements were conducted three times at room temperature with 1 mL water droplets. The surface roughness and morphology of the dimple nanostructures were measured via AFM (NX10, Park Systems, Suwon, Korea) in noncontact mode. The samples were coated onto a PET film by spray-coating.

## 3. Results and Discussion

### 3.1. Effect of the Synthesis Conditions on the Morphology of the CNF/Silica Nanocomposites

Figure 2 and Table 3 show the morphology of the CNF/silica nanocomposites obtained using different synthesis conditions—specifically, by varying the reaction solvent and catalyst concentrations. The size of the silica nanoparticles was determined via the TEM image analysis and the Zetasizer analysis, and the silica content was determined via TGA. In the CNF/silica-1, it was confirmed that anisotropic silica nanoparticles with a size of around 10–30 nm were adsorbed on the CNF surface (Figure 2a,b). Butler et al. reported that, under low [H_2_O]/[TEOS] molar ratios, a matrix with a more open structure was produced because of incomplete hydrolysis [41]. Increasing the water content of the CNF/silica nanocomposites (CNF/silica-2) resulted in an increase in the silica particle size and the formation of isotropic silica nanoparticles; additionally, the silica content increased from around 57.93 to 70.00 wt% (Figure 2c,d). Decreasing the ammonium hydroxide concentration of the CNF/silica-3 compared to CNF/silica-2 resulted in a decrease in the silica particle size and silica content to ~67.81 wt% (Figure 2e,f). The scale-up process (CNF/silica-4) produced similar results to CNF/silica-1. Indeed, the silica content decreased slightly because the ammonium hydroxide concentration was slightly lower, which can be expected to result in smaller silica nanoparticles. Figure 3 shows the morphology and schematic of the adsorption and separation of silica nanoparticles on the CNF surface. The adsorption of silica nanoparticles was investigated by measuring the zeta potential of CNFs and CNF/silica nanocomposite in ethanol solution at pH 12. The zeta potential of the CNFs and CNF/silica nanocomposite was −31.2 mV and −61.2 mV, respectively. This result shows that both CNFs and CNF/silica nanocomposites are in a stable state in an ethanol solution. In general, the threshold of stability of a colloidal nanoparticle solution in terms of the zeta potential is ±30 mV [40]. In particular, the CNFs have relatively lower repulsive force between nanofibers than silica nanoparticles. Therefore, the incorporation of silica nanoparticles in the CNFs makes the colloidal solution relatively more stable than CNFs alone. Previous studies have shown that silica nanoparticles are in a relatively more stable state at around −60 mV in a pH solution [42].

The addition of water with relatively strong hydrogen bonding results in the separation between silica nanoparticles and CNFs. It is considered that the affinity between CNFs and water is relatively higher than that of silica nanoparticles. Similarly, in other studies, it can be seen that CNFs have a high affinity for water, with a zeta potential of around −65 mV [43,44].

As a result, the adsorption of the silica nanoparticles on the CNF surface appears to be a phenomenon that occurs due to hydrogen bonding rather than covalent bonding. Additionally, the adsorbed silica nanoparticles were separated from the water solution using ultrasonic waves, and the silica nanoparticle size was measured using a Zetasizer. The size of silica nanoparticles in CNF/silica-1 was ~166 nm, while that in CNF/silica-4 was ~85 nm, showing rather large values, unlike TEM analysis. It is expected that anisotropic silica nanoparticles are actually agglomerated with one another. In particular, the difference in the silica nanoparticle size of CNF/silica-4 compared to CNF/silica-1 was thought to be due to the relatively high CNF concentration (0.20% to 0.33%) and low TEOS concentration (2.00% to 1.67%). As a result, the size and content of the silica nanoparticles can be controlled through water and ammonium hydroxide concentrations. In a previous work, it was found that increasing the water content resulted in an increase in the silica particle size and a decrease in the packing efficiency. Silica nanoparticles are generated and stabilized in an excess of an aqueous solution, which decreases their adsorption on the CNF surface [42].

### 3.2. Chemical Structure of the CNF, CNF/Silica, HDTMS-Modified CNF/Silica, and HDTMS-Modified CNF Nanocomposites

Figure 4 shows the FTIR spectra of the CNF, CNF/silica, HDTMS-modified CNF/silica, and HDTMS-modified CNF nanocomposites. The absorption peaks of the CNF/silica nanocomposites are attributed to the Si–OH and Si–O–Si vibration in the range 3100–3500 cm^−1^, as well as at 1080 and 800 cm^−1^ [42,45,46]. The absorption peaks at 3100–3500 and 1080 cm^−1^ partially overlap with the CNF absorption peak, and tend to shift toward slightly higher wavelengths. Modification with HDTMS yielded additional absorption peaks at 2925, 2845, and 1458 cm^−1^, which were attributed to the –CH_2_CH_2_ and –CH_3_ vibrations of the alkyl group (–C_16_H_33_) [36]. In Figure 4, the highlighted area represents the integral values of the spectra of the alkyl group (3038–2788 cm^−1^, and 1410–1400 cm^−1^) and the silane group (863–740 cm^−1^). Increasing the HDTMS concentration in CNF/silica nanocomposites increases the peak area of the alkyl group compared with that of the silane group. In h-sCNF 0.1, the peak of the alkyl group did not appear clearly, but the peak areas increased slightly from around 0.38 to 0.47 (3038–2788 cm^−1^) and from 0.10 to 0.15 (1419–1400 cm^−1^); therefore, it was inferred that a sufficient HDTMS modification had occurred. Similarly, the HDTMS modification of the CNF nanocomposites resulted in the appearance of peaks attributed to the alkyl group.

### 3.3. Characterization of the CNF, CNF/Silica, HDTMS-Modified CNF/Silica, and HDTMS-Modified CNF Nanocomposites

Table 4 and Figure 5 show the elemental composition of the CNF surface before and after modification with TEOS and HDTMS, as investigated via XPS. CNFs predominantly consist of C (56.20%) and O (43.80%), which is consistent with the (C_6_H_10_O_5_)_n_ composition of cellulose. The calculations of the amounts of CNFs, silica, and HDTMS are presented in the Appendix A. Here, the amount of HDTMS was calculated assuming CH_3_(CH_2_)_15_SiO_k_ (*k* = 1). In the CNF/silica nanocomposites, silica nanoparticles showed surface adsorption of ~55.15 wt% to the mass of the CNFs. As the concentration of HDTMS increased, the HDTMS modification of the CNF/silica nanocomposites increased from ~8.17 wt% to ~56.18 wt%, whereas the HDTMS modification of the CNF nanocomposites decreased from ~39.46 wt% to ~34.73 wt%. The results show a similar trend to the FTIR analysis.

As shown in Figure 5a–d, the high-resolution C1*s* spectra of the samples were fitted with three peaks—namely, C1, C2, and C3 at 284.3 ± 0.1, 286.0 ± 0.2, and 287.4 ± 0.3 eV, respectively, which correspond to the C–C/C–H/C–Si, C–O/C–OH, and O–C–O/C=O groups, respectively [47]. As shown in Figure 5e–k, the high-resolution Si2*p* spectra of the samples were fitted with three peaks—namely, S1, S2, and S3 at 102.0 ± 0.2, 102.9 ± 0.1, and 103.5 ± 0.1 eV, respectively, which correspond to the Si–C and SiO_2_ groups [47,48,49]. For Si2*p* spectra, the chemical structure was observed with binding energies of 102.9 eV and 103.5 eV corresponding to the Si oxidation states Si3^+^ and Si4^+^, respectively [48,50,51,52]. As shown in Figure 5a–d, the HDTMS modification of the CNF/silica nanocomposites resulted in an increase in the C1 peak (284.4 eV) and a significant decrease in the C2 (286.0 eV) and C3 (287.4 eV) peaks for oxygen-containing groups.

Similarly, the C1 peak is pronounced in h-CNF (Figure 5c,d), but it is difficult to find a trend for the HDTMS concentration. As shown in Figure 5e–h,j,k, the new peak corresponding to S1 (102.0 eV) gradually increases in the h-sCNF nanocomposites, thus showing a trend of increasing full width at half-maximum (FWHM) from 1.95 (CNF/silica) to 2.37. The h-CNF nanocomposites exhibit binding energy of S1 (102.0 eV) and S2 (102.9 eV) [48,49]; as a result, the appearance of new peaks corresponding to C1 (284.3 eV) and S1 (102.0 eV) indicates that the Si–C groups were introduced in the CNF and CNF/silica nanocomposites. In particular, the HDTMS modification at high concentrations for the CNF (h-CNF 0.8) nanocomposites results in a reduction in the S2 (103.0 eV) peak of SiO_2_ compared to the h-CNF 0.4 (Figure 5j,k); that is, in h-CNF 0.4, the SiO_2_ structure is considered to be relatively more dominant than in h-CNF 0.8. This part will be dealt with in more detail in the TGA and morphological analysis.

### 3.4. Thermal Degradation Properties of the CNF, CNF/Silica, HDTMS-Modified CNF/Silica, and HDTMS-Modified CNF Nanocomposites

Figure 6 shows the TGA and derivative thermogravimetric (DTG) curves of the CNF, CNF/silica, HDTMS-modified CNF/silica, and HDTMS-modified CNF nanocomposites. Table 5 summarizes the thermal degradation properties (*T_10_*, *T_inf, 1_*, *T_inf, 2_*, and *T_max_*) of the CNFs, silica, and inorganic/organic HDTMS. The inorganic/organic component content of HDTMS was calculated based on the results of the TGA and XPS analysis. The calculation is shown in detail in the Appendix A. The CNFs mainly exhibited first-order degradation (D_1st_) in the range from around 220 °C to 420 °C, and then almost degradation at a temperature exceeding 670 °C as second-order degradation (D_2nd_). The amorphous and crystalline components were found to be ~74.09 wt%, the total degradation of the crystalline cellulose polymer chain was ~25.03 wt%, and the remainder was ~0.88 wt% of ash. Such values are similar to those reported for the thermal decomposition of cellulose obtained from other plant sources [53], due to the different binding energies of the chemical structure and crystallinity [53,54,55]. The *T_10_* and *T_inf,1_* of the CNFs were around 311.23 °C and 346.87 °C, respectively. CNF/silica nanocomposites tended to increases to around 337.64 °C and 357.39 °C. The thermal stability of the silica nanoparticles was mainly due to the three-dimensional network structure of SiO_2_. Generally, the incorporation of inorganic materials into the polymer matrix can enhance thermal stability by acting as a superior insulator and mass transport barrier to the volatile products generated during decomposition [56,57]. Additionally, the volatilization of the CNFs may decrease because of the labyrinth effect of the silica nanoparticles [58,59]. This effect is expected to increase the ash content in the CNF nanocomposites from 0.88 wt% to 2.26 wt%. Similarly, the HDTMS modification gives rise to inorganic/organic HDTMS, and the inorganic composition serves to improve the thermal stability as an insulator does. In particular, HDTMS-modified CNF/silica nanocomposites have decreased inorganic content (silica and inorganic HDTMS) compared to CNF/silica, but increased *T_10_* and *T_inf,1_*, because organic HDTMS exhibits high thermal stability at a decomposition temperature from around 519 °C to 527 °C [60,61,62,63].

The HDTMS decomposed in a wide temperature range of ~200–500 °C, due to the relatively insufficient Si–O–Si crosslinking structure and the organic region of the hexadecyl group (–CH_2_(CH_2_)_14_CH_3_) [33,38,64,65]. Appendix A shows the differential scanning calorimetry (DSC) curve of the CNF/silica and HDTMS-modified CNF/silica nanocomposites. A melting temperature of ~44.76 °C was confirmed via the DSC analysis, proving that organic HDTMS was produced. Increasing the HDTMS concentration of the CNF/silica nanocomposites gradually increased the organic HDTMS content but slightly decreased the inorganic HDTMS content; therefore, the h-sCNF 1.0 and h-sCNF 2.0, which had relatively high organic content, exhibited a slight decrease in thermal stability compared to h-sCNF 0.5. However, increasing the HDTMS concentration of the CNF nanocomposites resulted in a decrease in the organic HDTMS content from around 26.00 wt% to 17.68 wt%. It can be seen that this is consistent with the trend of the previous FTIR and XPS analyses.

### 3.5. Surface Properties of the CNF, CNF/Silica, HDTMS-Modified CNF/Silica, and HDTMS-Modified CNF Nanocomposites

Figure 7 shows SEM and TEM images used to examine the modification of CNFs with HDTMS in more detail. Modification of the CNF nanocomposites at a low HDTMS concentration (h-CNF 0.4) results in the formation of HDTMS particles (Figure 7a,c). On the other hand, in h-CNF 0.8, a coating can be observed on the CNF surface, and the HDTMS particles are seldom observed (Figure 7b,d). It was expected that the HDTMS particles would be formed simultaneously with the coating of the CNF surface in the modification process. In particular, h-CNF 0.8 is probably due to the presence of a uniform coating on the CNF surface, which forms a hydrophobic surface, and the HDTMS particles are easily removed during the washing process. Therefore, h-CNF 0.8 shows a relatively lower HDTMS content than h-CNF 0.4 and, in particular, the organic HDTMS content is significantly reduced. These results are expected to show limitations in the modification with HDTMS due to the low surface area of the CNFs. Figure 8 shows the morphology of the HDTMS-modified CNF/silica nanocomposites with different HDTMS concentrations. Increasing the HDTMS concentration gradually increases the coating area on the CNF/silica nanocomposites. h-sCNF 2.0 is coated on the entire surface. Many studies have reported the HDTMS modification of the surface of silica nanoparticles. Chang et al. [35,36] reported that the silica nanoparticles were connected by a polymeric bridge, indicating a clear tendency to aggregate with one another. At a low HDTMS concentration, the surface of the silica nanoparticles became fuzzy, and tended to be coated with a thin layer. At a high HDTMS concentration, the silica particles were densely coated with the polymer to form a core–shell structure. Chang et al. [35] reported that an extremely dense polymer network was observed at an HDTMS concentration of 5%.

The WCAs were investigated to examine the hydrophobicity of the HDTMS-modified CNF nanocomposites. As shown in Figure 9a, the CNF and CNF/silica nanocomposites exhibited low water repellency, with a WCA of around 4.5° and 35.9°, respectively. The HDTMS-modified CNF nanocomposites exhibited water repellency, with a WCA of ~79.3°, and the HDTMS-modified CNF/silica nanocomposites showed high water repellency, with a WCA of ~159.1°. The HDTMS of h-CNF 0.4 and 0.8 was around 39.46 wt% and 34.73 wt%, respectively, whereas in h-sCNF 0.5 it was ~35.82 wt%, indicating high WCA despite similar or lower content. In particular, h-sCNF 2.0 exhibited superhydrophobicity, with a WCA of ~159.1°. This sample also exhibited superhydrophobicity in various aqueous solutions with different pH values (Figure 9b); this was mainly attributed to the hydrophobicity of the long alkyl chain of HDTMS. In addition, the incorporation of silica nanoparticles increased the roughness of the surface, thereby remarkably improving its water repellency.

As shown in Figure 10, the roughness values of each sample were examined via AFM. The CNF and HDTMS-modified CNF nanocomposites showed a smooth surface, with roughness values of around 49.67 nm, 36.65 nm, and 32.21 nm. The reason for the decreases in the roughness was that the roughness decreased due to the HDTMS coating on the CNF surface. On the other hand, the CNF/silica and HDTMS-modified CNF/silica nanocomposites showed relatively rough surfaces, with roughness values ranging from around 112.21 to 134.53 nm. These results show the effect that the incorporation of silica nanoparticles in the CNFs imparts nanoroughness. In addition, it was confirmed that the increase in HDTMS content imparted microroughness to HDTMS-modified CNF/silica nanocomposites by aggregating CNFs with one another (Figure 8). Similarly, many studies have reported the possibility to impart roughness and superhydrophobicity to the surface of silica particles using HDTMS [33,34,35,36,37,66]. Xu et al. [34] reported that the change in hydrophobicity was not significant when the HDTMS concentration was higher than 2%. Chang et al. [35] showed that a HDTMS concentration of 1 wt% is the most optimal. However, the hydrophobicity does not increase further with a higher HDTMS concentration, because the hierarchical structure features derived from the silica particles are hidden in the dense network of the coating; consequently, the roughness is no longer improved. Therefore, the HDTMS-modified CNF nanocomposites have limitations in terms of imparting hydrophobicity. On the other hand, the incorporation of silica nanoparticles into CNFs was more beneficial for the HDTMS modification; as a result, the content of organic HDTMS was increased. In addition, it is expected that a more pronounced surface modification of HDTMS may be possible due to the increased surface area.

## 4. Conclusions

Firstly, we controlled the adsorption behavior of silica nanoparticles on the surface of CNFs by adjusting the synthesis conditions. The silica nanoparticles’ size and the packing efficiency of the CNF surface could be controlled by varying the concentrations of ammonium hydroxide and water. Increasing the water and ammonium hydroxide concentrations increased the particle size of isotropic silica nanoparticles, while decreasing the concentration decreased the particle size of anisotropic silica nanoparticles. The incorporation of silica nanoparticles could impart thermal stability and surface roughness to the CNFs. The degradation temperature increased from around 311.23 °C and 346.87 °C to 337.64 °C and 357.39 °C, while the surface roughness increased from around 49.67 to 128.69. Above all, the phenomenon in which silica nanoparticles adsorbed on CNFs are desorbed again via a water-based solution has the potential to be applied to fields where recycling is easy.

Secondly, we successfully grafted HDTMS onto the surface of CNF and CNF/silica nanocomposites. The behavior of modified HDTMS was quantitatively analyzed via FTIR, XPS, TGA, SEM, and TEM. HDTMS modification could increase the thermal stability more effectively through the incorporation of silica nanoparticles, and imparted additional hydrophobicity. The HDTMS-modified CNF/silica nanocomposites could be controlled from around 88° to 159° through the HDTMS concentration. In particular, h-sCNF 2.0 exhibited superhydrophobicity, with a WCA of ~159.1° in various aqueous solutions with different pH values. This study can provide a reference for the adsorption of silica nanoparticles and the hydrophobic modification of CNF nanocomposites, as well as expand their fields of application.

## Figures and Tables

**Figure 1 polymers-14-00833-f001:**
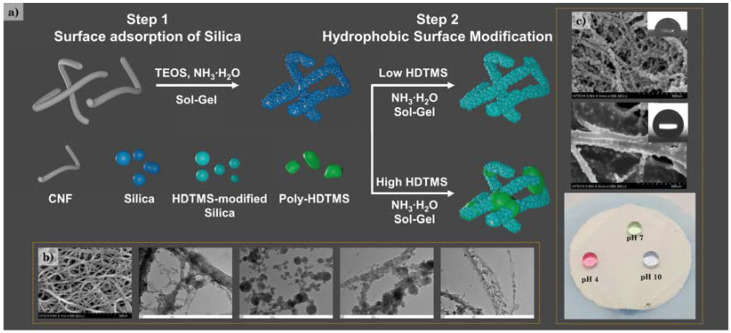
(**a**) Schematic of the preparation of HDTMS-modified CNF/silica nanocomposites, (**b**) morphology of CNFs and CNF/silica nanocomposites, and (**c**) morphology and water droplets on an HDTMS-modified CNF/silica nanocomposite with different pH values.

**Figure 2 polymers-14-00833-f002:**
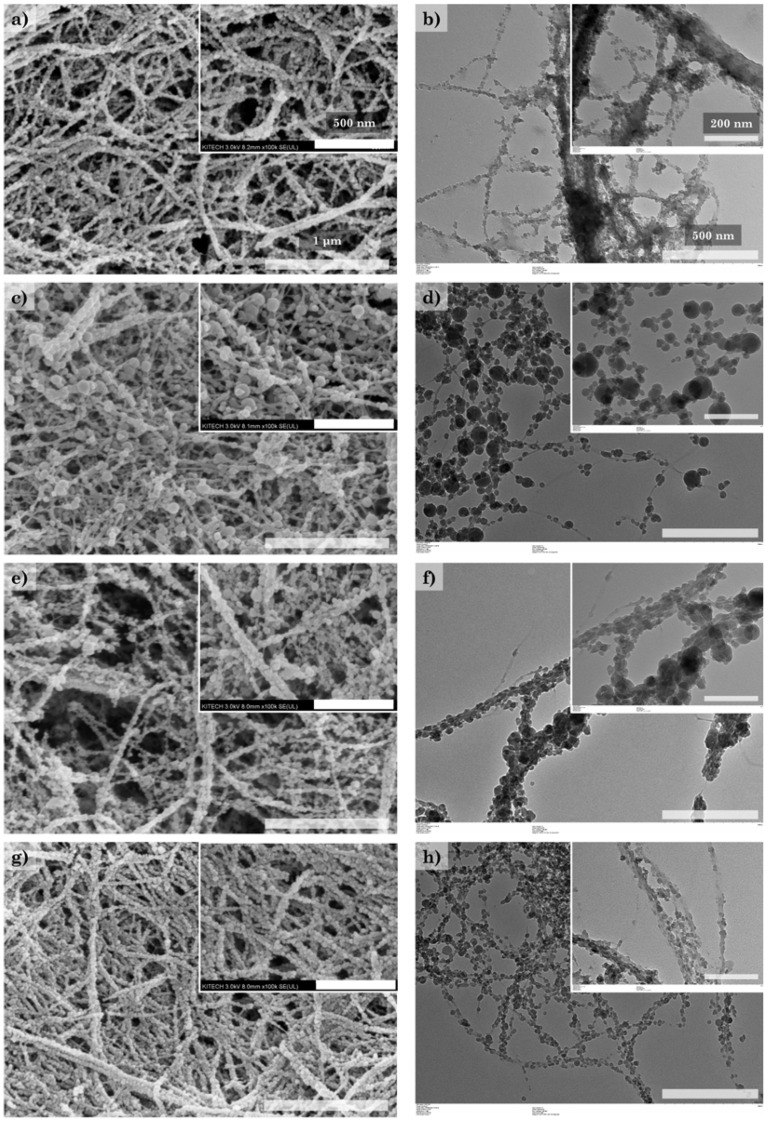
Morphology of CNF/silica nanocomposites investigated via SEM and TEM analysis: (**a**,**b**) CNF/silica-1, (**c**,**d**) CNF/silica-2, (**e**,**f**) CNF/silica-3, and (**g**,**h**) CNF/silica-4.

**Figure 3 polymers-14-00833-f003:**
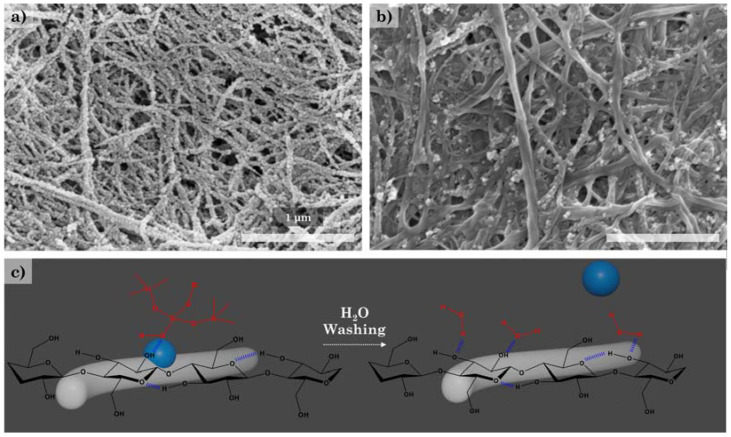
Morphology of the CNF/silica nanocomposites investigated via SEM analysis: (**a**) CNF/silica-1, (**b**) CNF/silica-1 after washing with water, and (**c**) scheme of adsorption and separation of silica nanoparticles on the CNFs’ surface.

**Figure 4 polymers-14-00833-f004:**
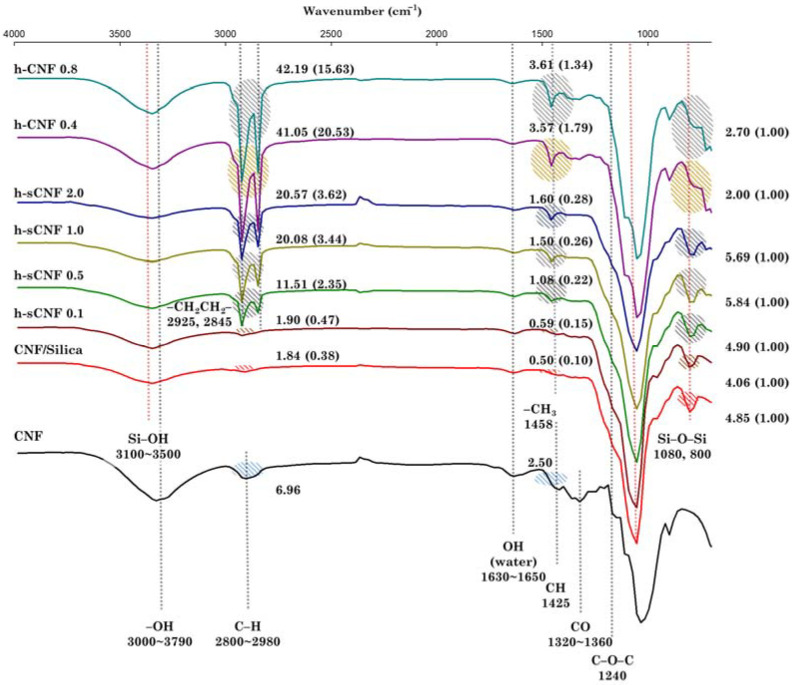
FTIR spectra of the CNF, CNF/silica, h-sCNF, and h-CNF nanocomposites in the range of 4000–600 cm^−1^, and measured absorption peak areas of the alkyl group (3038–2788, 1419–1400 cm^−1^) and the silane group (863–740 cm^−1^).

**Figure 5 polymers-14-00833-f005:**
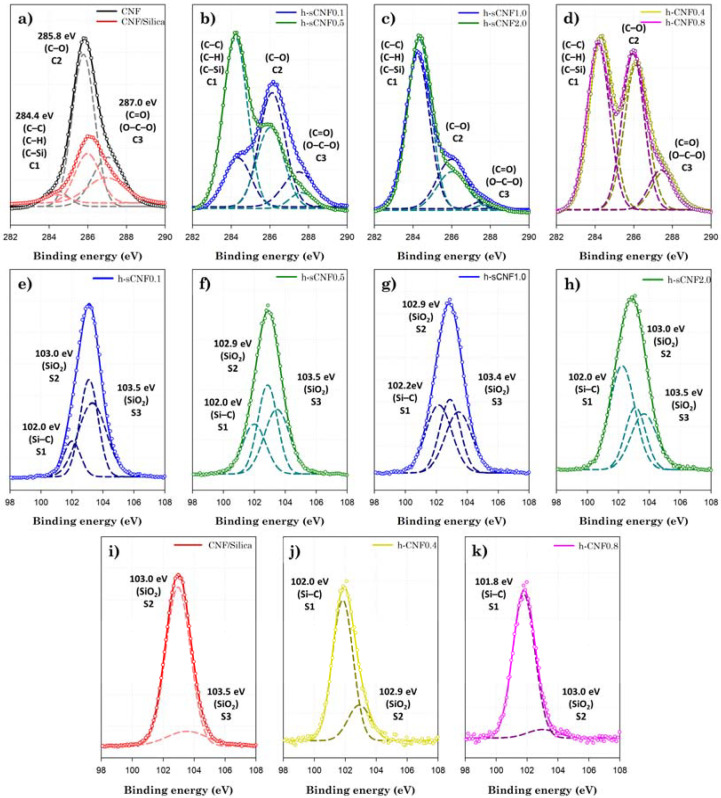
Chemical composition of the CNF, CNF/silica, h-sCNF, and h-CNF nanocomposites, as investigated via XPS: (**a**–**d**) high-resolution C1*s*, and (**e**–**k**) high-resolution Si 2*p* peak.

**Figure 6 polymers-14-00833-f006:**
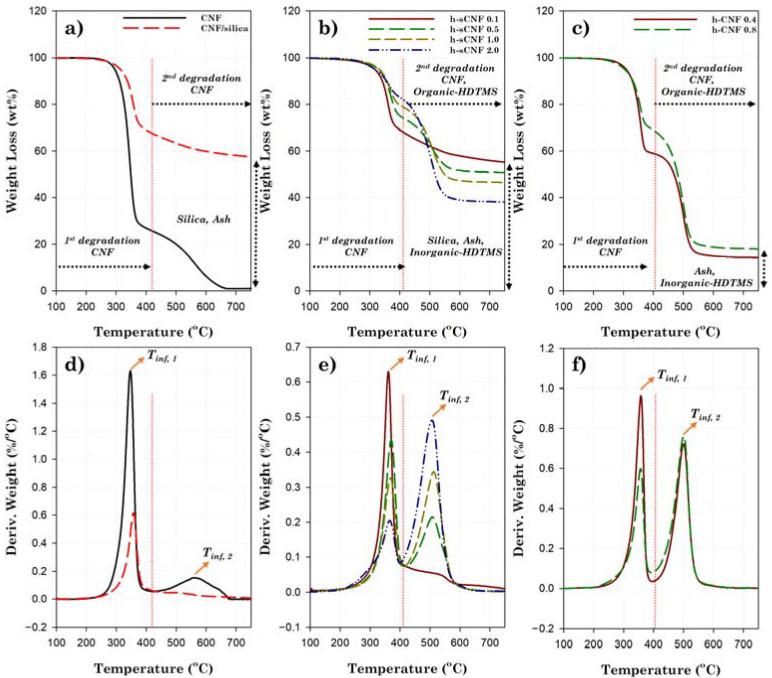
TGA and DTG curves of the CNF, CNF/silica, h-sCNF, and h-CNF nanocomposites: (**a**,**d**) CNF and CNF/silica, (**b**,**e**) h-sCNF, and (**c**,**f**) h-CNF nanocomposites.

**Figure 7 polymers-14-00833-f007:**
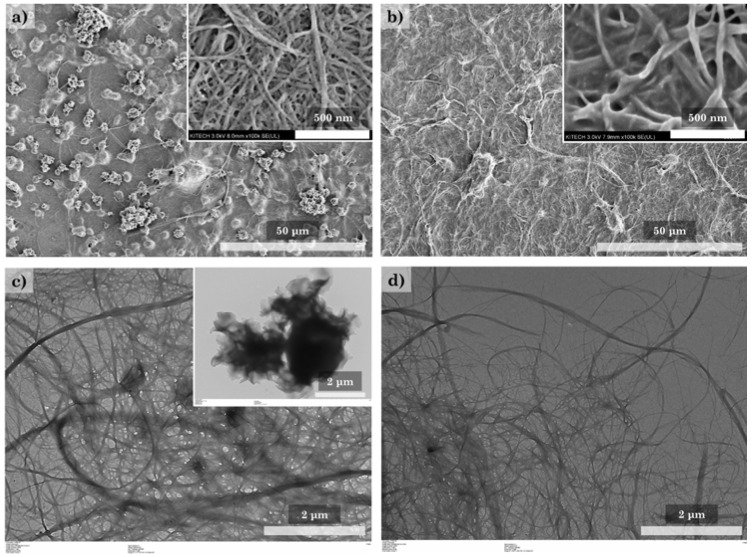
Morphology of the HDTMS-modified CNF nanocomposites, observed via SEM and TEM analysis: (**a**,**c**) h-CNF 0.4 and (**b**,**d**) h-CNF 0.8.

**Figure 8 polymers-14-00833-f008:**
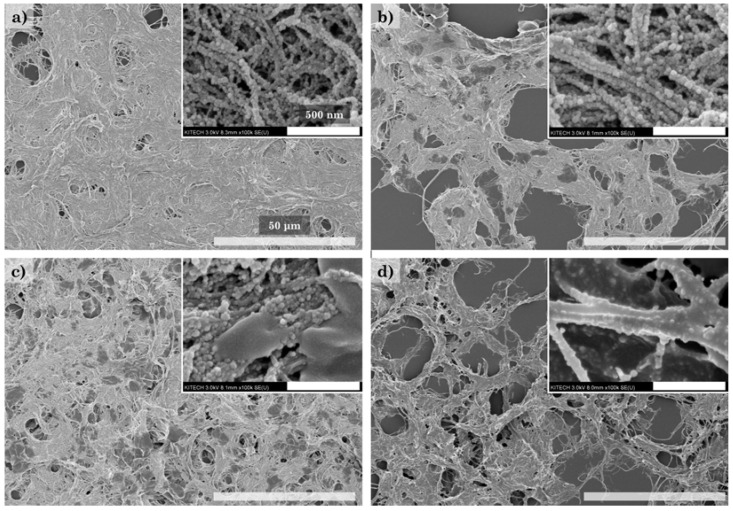
Morphology of the HDTMS-modified CNF/silica nanocomposites, observed via SEM analysis: (**a**) h-sCNF 0.1, (**b**) h-sCNF 0.5, (**c**) h-sCNF 1.0, and (**d**) h-sCNF 2.0.

**Figure 9 polymers-14-00833-f009:**
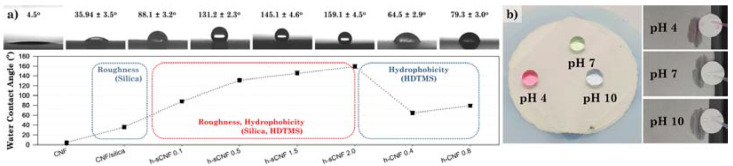
(**a**) Surface properties of the CNF, CNF/silica, h-sCNF, and h-CNF nanocomposites obtained via contact angle analysis. (**b**) Water droplets with pH values of 4, 7, and 10 exhibited a spherical shape on the surface.

**Figure 10 polymers-14-00833-f010:**
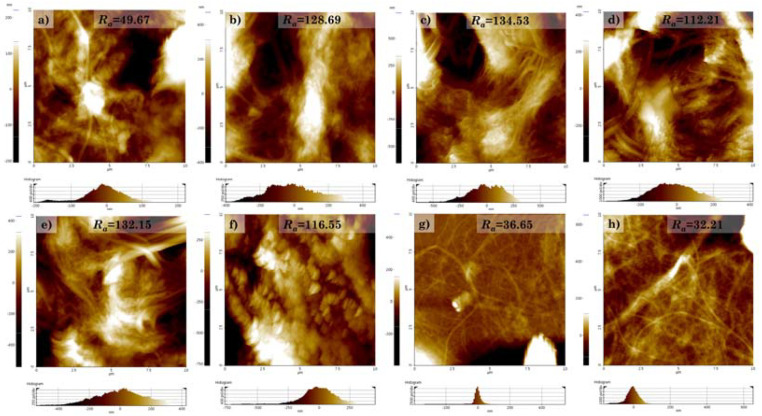
Surface roughness imaged via AFM (10 µm × 10 µm image): (**a**) CNF, (**b**) CNF/silica, (**c**) h-sCNF 0.1, (**d**) h-sCNF 0.5, (**e**) h-sCNF 1.0, (**f**) h-sCNF 2.0, (**g**) h-CNF 0.4, and (**h**) h-CNF 0.8.

**Table 1 polymers-14-00833-t001:** A typical recipe for the preparation of CNF/silica nanocomposites by varying the reaction solvent and catalyst concentrations.

Samples	CNF (g)	TEOS (mL)	EtOH (mL)	NH_4_OH (mL)	H_2_O (mL)	$\frac{\left[H_{2}O\right]}{\left[TEOS\right]}$	NH_4_OH (M)
CNF/silica-1	0.20	2.00	95.00	3.00	-	11.96	0.44
CNF/silica-2	0.20	2.00	85.50	3.00	9.50	70.39	0.44
CNF/silica-3	0.20	2.00	87.50	1.00	9.50	62.42	0.15
CNF/silica-4	0.80	4.00	230.00	6.00	-	11.96	0.36

**Table 2 polymers-14-00833-t002:** A typical recipe for the preparation of CNF/silica, h-CNF, and h-sCNF nanocomposites by varying the HDTMS concentrations.

Samples		CNF (g)	CNF/Silica (g)	TEOS (mL)	HDTMS (mL)	NH_4_OH (mL)	H_2_O (mL)
1 Step	CNF/silica	0.8	-	4.0	-	6.0	230.0
2 Step	h-sCNF 0.1	-	0.5	-	0.1	2.5	97.4
	h-sCNF 0.5	-	0.5	-	0.5	2.5	97.0
	h-sCNF 1.0	-	0.5	-	1.0	2.5	96.5
	h-sCNF 2.0	-	0.5	-	2.0	2.5	95.5
Control	h-CNF 0.4	0.2	-	-	0.4	2.5	97.1
	h-CNF 0.8	0.2	-	-	0.8	2.5	96.7

**Table 3 polymers-14-00833-t003:** Silica particle size and silica content of CNF/silica nanocomposites.

Samples	Particle Size		Silica Content (wt%) ^(c)^
	Size (nm) ^(a)^	Size (nm) ^(b)^	
CNF/silica-1	10 ~ 30	166.1	57.93
CNF/silica-2	13 ~ 90	96.57	70.00
CNF/silica-3	16 ~ 78	57.02	67.81
CNF/silica-4	10 ~ 30	85.07	57.67

^(a)^ Particle size measurement analysis of 30 samples by TEM images, (^b)^ by Zetasizer, and ^(c^) by TGA analysis.

**Table 4 polymers-14-00833-t004:** Surface composition of the CNF, CNF/silica, h-sCNF, and h-CNF nanocomposites, as measured via XPS.

Samples	Atomic Percentage (%)			Amount (%)			FWHM
	C	O	Si	CNF	Silica	HDTMS	
CNF	56.20	43.80	-	100.00	-	-	-
CNF/silica	29.20	54.65	16.15	44.85	55.15	-	1.96
h-sCNF 0.1	33.94	50.97	15.05	40.65	51.18	8.17	1.98
h-sCNF 0.5	52.82	35.78	11.40	30.14	34.04	35.82	2.11
h-sCNF 1.0	57.14	31.85	11.01	24.98	31.09	43.92	2.18
h-sCNF 2.0	64.82	25.66	9.52	20.29	23.53	56.18	2.37
h-CNF 0.4	67.30	30.55	2.15	60.53	-	39.46	1.82
h-CNF 0.8	66.71	31.37	1.92	65.27	-	34.73	1.76

**Table 5 polymers-14-00833-t005:** Thermal degradation properties (*T_10_*, *T_inf,1_*_,_
*T_inf,2_*_,_ and *T_max_*) of the CNF, CNF/silica, h-sCNF, and h-CNF nanocomposites, and inorganic/organic component content of HDTMS for the h-CNF and h-sCNF nanocomposites.

Samples	*T_10_* (°C)	*^a^ T_inf, 1_* (°C)	*^a^ T_inf, 2_* (°C)	*^b^ T_max_* (wt%)	Inorganic HDTMS (wt%)	Organic HDTMS (wt%)
CNF	311.23	346.87	560.01	0.88	-	-
CNF/silica	337.64	357.39	-	57.41	-	-
h-sCNF 0.1	341.63	360.87	-	55.02	1.58	6.59
h-sCNF 0.5	353.85	370.49	508.44	50.77	14.47	21.35
h-sCNF 1.0	357.16	367.86	512.58	46.54	13.19	30.73
h-sCNF 2.0	355.69	365.05	506.00	38.03	12.24	43.94
h-CNF 0.4	329.88	357.91	500.05	14.34	13.46	26.00
h-CNF 0.8	331.74	356.59	500.82	17.93	17.05	17.68

^*a*^: Inflection point of the TGA curve; ^*b*^: residual amount.

## Data Availability

Not applicable.

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
