# Peer review of "Fabrication and Characterization of Hydrophobic Cellulose Nanofibrils/Silica Nanocomposites with Hexadecyltrimethoxysilane"

_polymers, 2022, doi:10.3390/polym14040833_

Round 1

Reviewer 1 Report

This manuscript deals with the preparation of HDTMS-modified CNF/silica nanocomposites with superhydrophobic properties. The subject is interesting, the paper content is clearly presented. Nevertheless, some revisions are needed before its publications.

  • The introduction is poor, and the novelty was not enough remarked. Additional info should be included to differentiate this work from the already reported in previous manuscripts.
  • There is no evidence why authors are using HDTMS instead other similar compounds. What are their main advantages?
  • Author mentioned that the applications of nanocellulose are limited due to the lack of thermoplasticity and hydrophobicity. However, there is no info related to the applications of the HDTMS-modified CNF/silica nanocomposites prepared in this work. In case of using in sectors such as agrifood, biomedical and so on, what is the toxicity and biocompatibility of HDTMS?
  • Authors mention that they control the adsorption behavior of silica nanoparticles on the surface of CNF by adjusting the synthesis conditions (by varying the ammonium hydroxide and water concentrations.). In this sense I miss a complementary technique to monitor this absorption as well as the surface modification of CNF upon treatment with silica nanoparticles. For instance: Zeta Potential, quartz crystal microgravimetry or something similar
  • What is the meaning of Bio-TEM analysis? What was the difference with TEM analysis?
  • Silica nanoparticles are appreciated in figure 2f. However, it seems that CNF are missing. Authors should change this micrograph
  • Details images of figures 6 are not shown clearly.
  • AFM images and profiles of figure 7 are not shown clearly
  • Conclusions are too much summarized. Authors should give more details. Additionally they should repeat in this section the composition of this sample since it is the only one which exhibited superhydrophobicity.

Author Response

Response to Reviewer 1 Comments

Point 1: The introduction is poor, and the novelty was not enough remarked. Additional info should be included to differentiate this work from the already reported in previous manuscripts.

There is no evidence why authors are using HDTMS instead other similar compounds. What are their main advantages?

Response 1: Thank you for the kind comment.

We have added a description and references.

In Page 2, line 63-64: Nevertheless, fluorinated compounds are expensive and may accumulate and become toxic in organisms and the environment.[32,33]

In page 2, 73-76: It was attempted to impart hydrophobicity using HDTMS having along alkyl group, and precursors such as MTES having a relatively short alkyl group were excluded from the selection because the formation of nanoparticles was dominant.

In page 2, 81-86: The HDTMS-modified CNF/silica nanocomposite could be controlled from about 88° to 159° through the HDTMS concentration (with a HDTMS concentration of about 0.1 to 2.0 wt%). In addition, as a result of the hydrolysis-condensation reaction of HDTMS, it was possible to obtain the contents of the thermally decomposed organic region and the non-decomposed inorganic region through TGA.

Point 2: Author mentioned that the applications of nanocellulose are limited due to the lack of thermoplasticity and hydrophobicity. However, there is no info related to the applications of the HDTMS-modified CNF/silica nanocomposites prepared in this work. In case of using in sectors such as agrifood, biomedical and so on, what is the toxicity and biocompatibility of HDTMS?

Response 2: Deleted the content about thermoplasticity

In page 1: The applications of nanocellulose are limited due to the lack of hydrophobicity, which is caused by the presence of a large number of hydroxyl groups (–OH).

The toxicity and biocompatibility of HDTMS are still unknown. Silica and silicates are generally recognized to be safe as oral delivery ingredients in amounts up to 1500 mg per day. doi.org/10.1016/j.cis.2017.04.005

Point 3: Authors mention that they control the adsorption behavior of silica nanoparticles on the surface of CNF by adjusting the synthesis conditions (by varying the ammonium hydroxide and water concentrations.). In this sense I miss a complementary technique to monitor this absorption as well as the surface modification of CNF upon treatment with silica nanoparticles. For instance: Zeta Potential, quartz crystal microgravimetry, or something similar

Response 3: We have added a description and references.

In page 4, 146-153:

2.4 Zetasizer Analyzer and Zeta potential analysis

The Z-average particle size was measured at 25 °C with a zetasizer (Nano ZS, Malvern Instruments Ltd.) using a ~ 0.1% dispersion solution. The zeta potential (ζ) was determined using a laser electrophoresis zeta potential analyzer (Nano ZS, Malern Instruments Ltd.). The zeta potential analysis was performed at pH 12 at 25 °C. The zeta potential was calculated from the electrophoretic mobility (µe) using Smoluchowski’s equation,

where ζ is the zeta potential in mV, ε is the dielectric constant of the medium, η is the viscosity of solution, and μ_e is the electrohpretic mobility.[41]

In page 5, 191-200: The adsorption of silica nanoparticles was investigated by measuring the zeta potential of CNF and CNF/silica nanocomposite in ethanol solution at pH 12. The zeta potential of the CNF and CNF/silica nanocomposite are -31.2 mV and -61.2 mV, respectively. This result shows that both CNF and CNF/silica nanocomposites are in a stable state in an ethanol solution. In general, the threshold of stability of a colloidal nanoparticle solution in terms of the zeta potential is ± 30 mV.[41] In particular, The CNF has relatively lower repulsive force between nanofibers than silica nanoparticles. Therefore, the incorporation of silica nanoparticles on the CNF makes the colloidal solution relatively more stable than of CNF. Previous studies have shown that silica nanoparticles are in a relatively more stable state at about -60 mV in a pH solution. [43]

Point 4: What is the meaning of Bio-TEM analysis? What was the difference with TEM analysis?

Response 4: We have corrected from Bio-TEM to TEM.

Point 5: Silica nanoparticles are appreciated in figure 2f. However, it seems that CNF are missing. Authors should change this micrograph.

Response 5: We have corrected in figure 2.

In page 6, 7 : Figure 2 was modified by dividing it into Figure 2 and Figure 3.

Point 6: Details images of Figures 6 are not shown clearly.

Response 6: We have corrected in Figure 6.

In page 12-13: Figure 6 was modified by dividing it into Figure 7 and Figure 8.

Point 7: AFM images and profiles of Figure 7 are not shown clearly

Response 7: We have corrected in Figure 7.

In page 14: Figure 7 was modified to Figure 10.

Point 8: Conclusions are too much summarized. Authors should give more details. Additionally they should repeat in this section the composition of this sample since it is the only one which exhibited superhydrophobicity.

Response 8: We have added a description

In page 14, 422-436: Increasing the water and ammonium hydroxide concentration increases the formation and particle size of isotropic silica nanoparticles, while decreasing the concentration de-creases the anisotripic silica nanoparticles and particle size. The incorporation of silica nanoparticles could impart thermal stability and surface roughness to the CNF. The degradation temperature increased from about 311.23 °C and 346.87 °C to 337.64 °C and 357.39 °C, and the surface roughness increased from about 49.67 to 128.69.

In page 15, 436: The HDTMS-modified CNF/silica nanocomposites could be controlled from about 88° to 159° through the HDTMS concentration.

Thank you for the kind comment. All comments are addressed in the manuscript.

Reviewer 2 Report

This manuscript reports that to increase the hydrophobicity of CNF, silica nanoparticles and HDTM are successfully introduced into CNF. The silica nanoparticle size and packing efficiency on the CNF surface could be controlled by varying the ammonium hydroxide and water concentrations. In addition, hexadecyltrimethoxysilane (HDTMS) was successfully grafted onto CNF or CNF/silica nanocomposite surfaces, and quantitative content of organic/inorganic substances in HDTMS was analyzed through XPS and TGA.

This manuscript displayed reasonable conclusion. Therefore, I agree that this manuscript can be published in polymers. However, some format modification and data discussion need to be provided before it can be published. The specific reasons are shown below:

  1. What is the specific definition of organic/inorganic HDTMS, please give a clear explanation.
  2. What is the relationship between thermal stability and organic/inorganic HDTMs, it is better to cite some references to explain.
  3. Fig4 should use a centered format, which is consistent with other Figs.
  4. The incorporation of silica nanoparticles could serve to prevent agglomeration in nanocellulose during drying in line 381. Please give an explanation and explanation in Results and Discussion section.
  5. The year on line 415 in the references has no boldface format.
  6. The authors could add the following references which would again increase the interest to general functional cellulosic material readers: Journal of Bioresources and Bioproducts, 2021, 6(1): 26-32; ACS Applied Materials & Interfaces, 2021, 13, 7617-7624; Journal of Bioresources and Bioproducts, 2021, 6(1): 75-81.

Author Response

Response to Reviewer 2 Comments

Point 1: What is the specific definition of organic/inorganic HDTMS, please give a clear explanation.

Response 1: Thank you for the kind comment.

We have added a description and references.

In Page 2, line 83-85: In addition, as a result of the hydrolysis-condensation reaction of HDTMS, it was possible to obtain the contents of the thermally decomposed organic region and the non-decomposed inorganic region through TGA.

Point 2: What is the relationship between thermal stability and organic/inorganic HDTMs, it is better to cite some references to explain.

Response 2: We have added a description and references.

In page 10, line 320-322: The reason is that organic HDTMS exhibits high thermal stability at a decomposition temperature of about 519°C to 527°C.[61-64]

Point 3: Fig4 should use a centered format, which is consistent with other Figs.

Response 3: We have corrected in Figure 4.

Point 4: The incorporation of silica nanoparticles could serve to prevent agglomeration in nanocellulose during drying in line 381. Please give an explanation and explanation in Results and Discussion section.

Response 4: The part that prevents aggregation has been deleted.

Point 5: The year on line 415 in the references has no boldface format.

Response 5: We have corrected.

Point 6: The authors could add the following references which would again increase the interest to general functional cellulosic material readers: Journal of Bioresources and Bioproducts, 2021, 6(1): 26-32; ACS Applied Materials & Interfaces, 2021, 13, 7617-7624; Journal of Bioresources and Bioproducts, 2021, 6(1): 75-81.

Response 6: We have added a description and references.

In page 2, line 57: Added references 29, 30, and 31.

Thank you for the kind comment. All comments are addressed in the manuscript.
